# Method for cycle detection in sparse, irregularly sampled, long-term neuro-behavioral timeseries: Basis pursuit denoising with polynomial detrending of long-term, inter-ictal epileptiform activity

Irena Balzekas[1,2,3,4]*, Joshua Trzasko[5], Grace Yu[3,4], Thomas J. Richner[1], Filip Mivalt[1,6,7], Vladimir Sladky[1,6,8], Nicholas M. Gregg[1], Jamie Van Gompel[9], Kai Miller[9], Paul E. Croarkin[10], Vaclav Kremen[1,11], Gregory A. Worrell[1,2]*

1 Bioelectronics, Neurophysiology, and Engineering Laboratory, Department of Neurology, Mayo Clinic, Rochester, Minnesota, United States of America, 2 Biomedical Engineering and Physiology Graduate Program, Mayo Clinic Graduate School of Biomedical Sciences, Rochester, Minnesota, United States of America, 3 Mayo Clinic Alix School of Medicine, Rochester, Minnesota, United States of America, 4 Mayo Clinic Medical Scientist Training Program, Rochester, Minnesota, United States of America, 5 Department of Radiology, Mayo Clinic, Rochester, Minnesota, United States of America, 6 International Clinic Research Center, St. Anne's University Research Hospital, Brno, Czech Republic, 7 Faculty of Electrical Engineering and Communication, Department of Biomedical Engineering, Brno University of Technology, Brno, Czechia, 8 Faculty of Biomedical Engineering, Czech Technical University in Prague, Czechia, 9 Department of Neurosurgery, Mayo Clinic, Rochester, Minnesota, United States of America, 10 Department of Psychiatry and Psychology, Mayo Clinic, Rochester, Minnesota, United States of America, 11 Czech Institute of Informatics, Robotics and Cybernetics, Czech Technical University in Prague, Prague, Czechia

* balzekas.irena@mayo.edu (IB); worrell.gregory@mayo.edu (GAW)

**Data Availability Statement:** Analyses were run on a computer with Intel Xeon Silver 4108 CPU 1.80

## Abstract

Numerous physiological processes are cyclical, but sampling these processes densely enough to perform frequency decomposition and subsequent analyses can be challenging. Mathematical approaches for decomposition and reconstruction of sparsely and irregularly sampled signals are well established but have been under-utilized in physiological applications. We developed a basis pursuit denoising with polynomial detrending (BPWP) model that recovers oscillations and trends from sparse and irregularly sampled timeseries. We validated this model on a unique dataset of long-term inter-ictal epileptiform discharge (IED) rates from human hippocampus recorded with a novel investigational device with continuous local field potential sensing. IED rates have well established circadian and multiday cycles related to sleep, wakefulness, and seizure clusters. Given sparse and irregular samples of IED rates from multi-month intracranial EEG recordings from ambulatory humans, we used BPWP to compute narrowband spectral power and polynomial trend coefficients and identify IED rate cycles in three subjects. In select cases, we propose that random and irregular sampling may be leveraged for frequency decomposition of physiological signals.

**Trial Registration:** NCT03946618.

 COMPUTATIONAL BIOLOGY

Cycle detection in sparse epileptiform activity rate timeseries

GHz, 188 GB RAM, 16 physical cores, and 32 logical cores, and running Ubuntu version 18.04.6. Code, deidentified raw data, and simulated data are available on GitHub at (https://github.com/irenabalzekas/BPWP).

**Funding:** This research was supported by the National Institutes of Health National Institute of Neurological Disorders and Stroke (https://www.ninds.nih.gov/) grants UH2/UH3-NS95495 and R01-NS09288203 (Principal investigator GAW). IB received support from MSTP grant T32GM145408. VK was partially supported by institutional resources from Czech Technical University in Prague, Czech Republic. The funders had no role in study design, data collection and analysis, decision to publish, or preparation of the manuscript.

**Competing interests:** I have read the journal's policy and the authors of the manuscript have the following competing interests: IB has received compensation from an internship with Cadence Neuroscience Inc., for work unrelated to the current publication. FM has received salary support from Cadence Neuroscience Inc. PEC has received research grant support from Neuronetics, Inc.; NeoSync, Inc; and Pfizer, Inc. PEC has received grant-in-kind (equipment support for investigator-initiated research studies) from Assurex; MagVenture, Inc; and Neuronetics, Inc. He has served on advisory boards for Engrail Therapeutics, Myriad Neuroscience, Procter & Gamble, Sunovion, and Meta Platforms. JVG, GAW, BNL, and BHB are named inventors for intellectual property licensed to Cadence Neuroscience Inc. BNL, JVG, GAW, and NG are investigators for the Medtronic EPAS trial, SLATE trial, and Mayo Clinic Medtronic NIH Public Private Partnership (UH3-NS95495), also with consulting contract. JVG and GAW own stock and have consulting contracts with Neuro-One Inc. JVG is the site primary investigator in the Polyganics ENCASE II trial, NXDC Gleolan Men301 trial, and the Insightec MRgUS EP001 trail. JT has royalty bearing intellectual property agreements with General Electric Healthcare, Shenzhen Mindray Bio-Electric Corp, and Sonoscape Medical Corp. NMG is consulting for NeuroOne (money to Mayo Clinic).

## Author summary

Circadian and multiday cycles are an important part of many long-term neuro-behavioral phenomena such as pathological inter-ictal epileptiform discharges (IEDs) and seizures in epilepsy. Long-term, ambulatory, neuro-behavioral monitoring in human patients involves complex recording systems that can be subject to intermittent, irregular data loss and storage limitations, resulting in sparse, irregularly sampled data. Cycle identification in sparse data or irregular data using traditional frequency decomposition techniques typically requires interpolation to create a regular timeseries. Using unique, long-term recordings of pathological brain activity in people with epilepsy implanted with an investigational device, we developed a method to identify cycles in sparse, irregular neuro-behavioral data without interpolation. We anticipate this approach will enable retrospective cycle identification in sparse neuro-behavioral timeseries and support prospective sparse sampling in monitoring systems to enable long-term monitoring of patients and to extend storage capacity in a variety of ambulatory monitoring applications.

## Introduction

Studying the dynamics of complex neurological and psychiatric diseases with behavioral manifestations requires ambulatory monitoring of peripheral physiology, behavior, and brain activity. In epilepsy, long-term monitoring of seizures and inter-ictal epileptiform discharges (IEDs) has revealed circadian and multiday cycles of seizures and epileptiform brain activity [1–3]. In temporal lobe epilepsy (TLE) seizures and IED rates are especially tied to sleep/wake behavioral states [3–7]. Leveraging these cycles to anticipate high risk intervals for seizure occurrence and time interventions may create new therapeutic opportunities [3].

Data under-sampling and loss are common in ambulatory monitoring, especially for devices recording and streaming intracranial electroencephalography (iEEG). Complex recording systems (Fig 1A) are subject to signal storage, battery life, and wireless connectivity limitations [8]. Given device data storage limitations, most electrical brain stimulation (EBS) devices in clinical use store variable and limited local field potential (LFP) data: around one month of coarsely averaged features such as power-in-band, for example [9,10] or limited brief selected raw data segments [9,10]. Multiday data drops tend to present the largest analytical challenges. In loop recording devices, often the past data are overwritten if recordings are not downloaded by patients [9]. Research grade, investigational devices capable of continuous data streaming require frequent charging and consistent wireless connectivity; creating significant hardware management burdens for patients [8,11,12]. Power and connectivity issues lead to packet drops and data loss that can degrade analytics and clinical performance [13]. Approaches that maximize the collection of meaningful data while minimizing demands on patients and system hardware are needed.

One solution is to store only the minimum data required to characterize a neurophysiological process [14]. An acceptable, minimal representation of data depends on the signal of interest and requirements of methods for analysis [15]. Most approaches for cycle identification are unsuitable for infrequently and irregularly sampled timeseries. Discrete Fourier-[16] and auto-correlation-based [17] linear methods require dense and regular sampling, increasing storage demands and necessitating interpolation of missing data. Identifying cycles in timeseries wherein most samples are unknown requires approaches designed to accommodate sparsity.

Sparsity can describe datasets, signals, and statistical approximations. These definitions are often concurrently relevant. In a sparse dataset, most observations, or samples, are zero.

## A. Use case: Continuous neuro-behavioral recordings

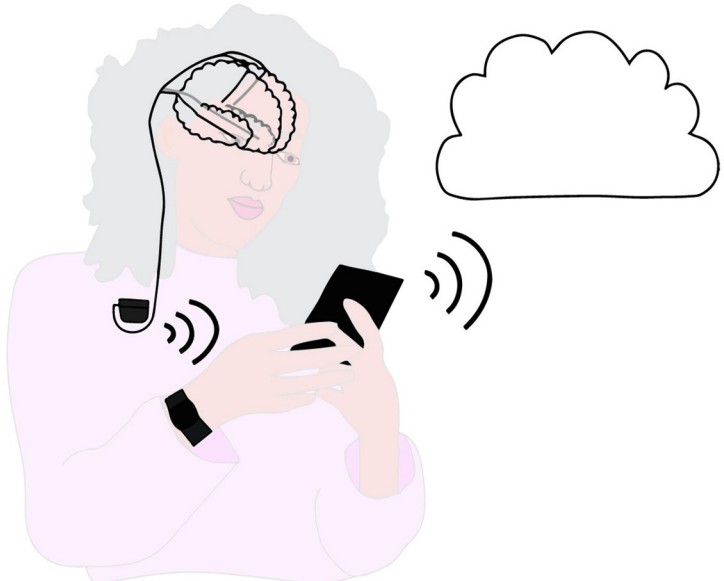

## B. Application: Inter-ictal epileptiform activity

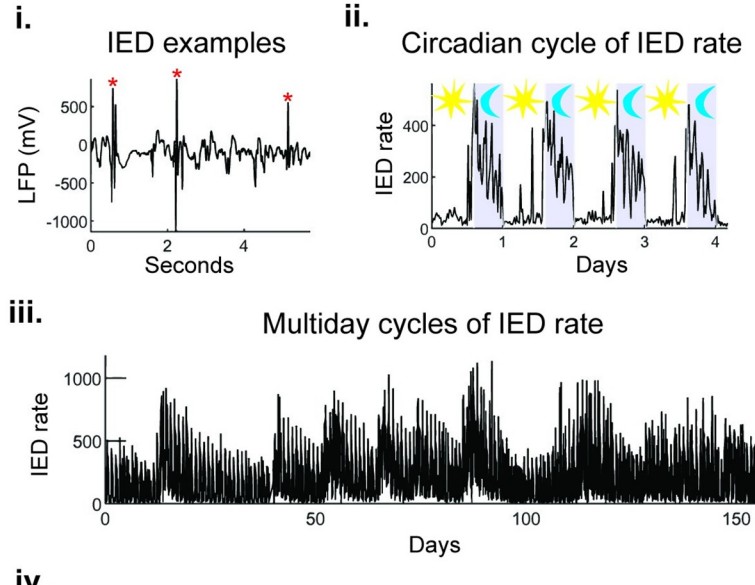

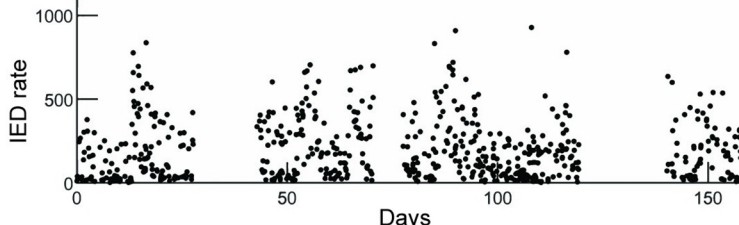

**Fig 1. Schematic overview of neuro-behavioral recording applications.** (A) Use case: Systems for continuous neuro-behavioral recordings. Recording, energy, storage, connectivity, and usability demands placed on ambulatory brain recording systems can result in data loss. (B) Application: Cyclical neuro-behavioral signals–Inter-ictal epileptiform activity. (i) Representative example of inter-ictal epileptiform discharges (IEDs) recorded from intracranial electroencephalography (iEEG). (ii) Example of IED rates (IED per hour) recorded from a hippocampal iEEG electrode showing circadian changes in IED rate with increased IED rates during sleep. The period from 10 PM to 8 AM is noted in gray. (iii) Example of multiday IED rate recording showing multiday cycles. (iv) Random subsampling of the signal in iii to show how sparse sampling and data drops make it challenging to discern the underlying cycles of IED rate.

Sometimes, a signal can be well represented by a small number of components (low rank approximation) that capture nearly all the variance of a signal. In this case the signal, and its model, are both sparse. There are mathematical definitions whereby values describe the number of nonzero coefficients. For example, a signal is k-sparse if only k coefficients are nonzero. In sparse representations, the value of k is small, though its exact value is case dependent. This can also be explained as sparsity in a transform domain. For example, although a signal may have been sampled at a high sampling rate to create a very dense dataset in the time domain, in the frequency domain, the power spectrum may only have 3, narrow peaks indicating that what was observed was 3, separable, concurrent oscillations. Altogether, a signal that has a few, distinctive, separable components that capture most of the signal's variance lends itself well to sparse representation and ultimately analysis or reconstruction of a sparse dataset using sparsity- (or low-rank)-based techniques.

Sparse recovery techniques (including compressed sensing methods) capture a family of techniques developed to recover information from sparse and sparsely sampled data in applications including signal compression and magnetic resonance imaging (MRI)[18–20]. Based on basis pursuit [21], these methods exploit conditions whereby algebraically undetermined or sub-Nyquist-Shannon sampling is permissible and frequency decomposition can be done on less frequently sampled signals [19]. The related techniques of basis pursuit denoising and least absolute shrinkage selection operator (LASSO), also known as L1 regularization or sparse regression, estimate the simplest, or in this case the sparsest, description for a set of observations that does not have to be regularly or densely sampled [22,23]. By leveraging sparsity of the target signal quantity, we can address the challenge presented by long-term neuro-behavioral data acquisition: identifying cycles in infrequently and irregularly sampled data [24]. Our goal was to develop a conservative model for frequency decomposition of infrequently and irregularly sampled data that yields sparse spectral outputs and accommodates polynomial trends in the timeseries, limiting potential errors introduced by linear detrending during pre-processing.

Here, we describe a method for frequency decomposition of sparsely and irregularly sampled neuro-behavioral data: basis pursuit denoising with polynomial detrending (BPWP). In short, the approach takes an irregularly and sparsely sampled timeseries and performs frequency decomposition and polynomial detrending. We used simulations to test the method's characterization of cycles in complex timeseries data over a range of physiologically relevant parameters. We then applied BPWP to unique, real-world recordings of chronic, ambulatory, iEEG from people implanted with the investigational Medtronic Summit RC+S deep brain stimulation (DBS) device for drug resistant temporal lobe epilepsy (TLE). The recordings contained multiple years of translationally relevant IED rates from epileptic hippocampus in 3 human participants with mesial temporal lobe epilepsy in an investigational device study (https://clinicaltrials.gov/study/NCT03946618). We identified circadian and multiday cycles of IED rate in our data with both standard approaches and BPWP. Our findings support frequency decomposition of sparsely sampled neuro-behavioral data and highlight translational opportunities for efficient sampling and reconstruction of neuro-behavioral signals to study the interplay between electrophysiology and behavioral or multi-domain data.

## Results

The purpose of BPWP is to identify oscillations and polynomial trends in sparsely and irregularly sampled timeseries. A schematic diagram of the approach is provided in Fig 2. Explained in detail in the methods, the model assumes the observed signal (a set of sparse, irregular samples) can be described by the linear sum of a set of oscillations, a polynomial trend, and noise. BPWP aims to minimize the square of the L2 norm of the difference between the observed data and the estimate of the observed data (the sum of the oscillations, polynomial trend, and noise). The model outputs consist of a sparse, narrowband power spectrum (spectral coefficients mapping to discrete cosine transform (DCT) basis) and a polynomial trend. BPWP constrains the number and amplitude of the spectral coefficients by restricting the L1 norm of the coefficient vector to be less than the parameter $\delta$. The $\delta$ parameter is selected in a subject-specific manner via parameter sweeps to identify the $\delta$ value that minimizes the mean square error between the observed data and the model estimates. The spectrum and polynomial trend are then used to estimate the regular, dense representation of the signal that yielded the sparse, irregular samples. We first demonstrate the method on a simulated IED-rate timeseries with pre-defined spectral composition. We then test the method on real-world IED rate data from

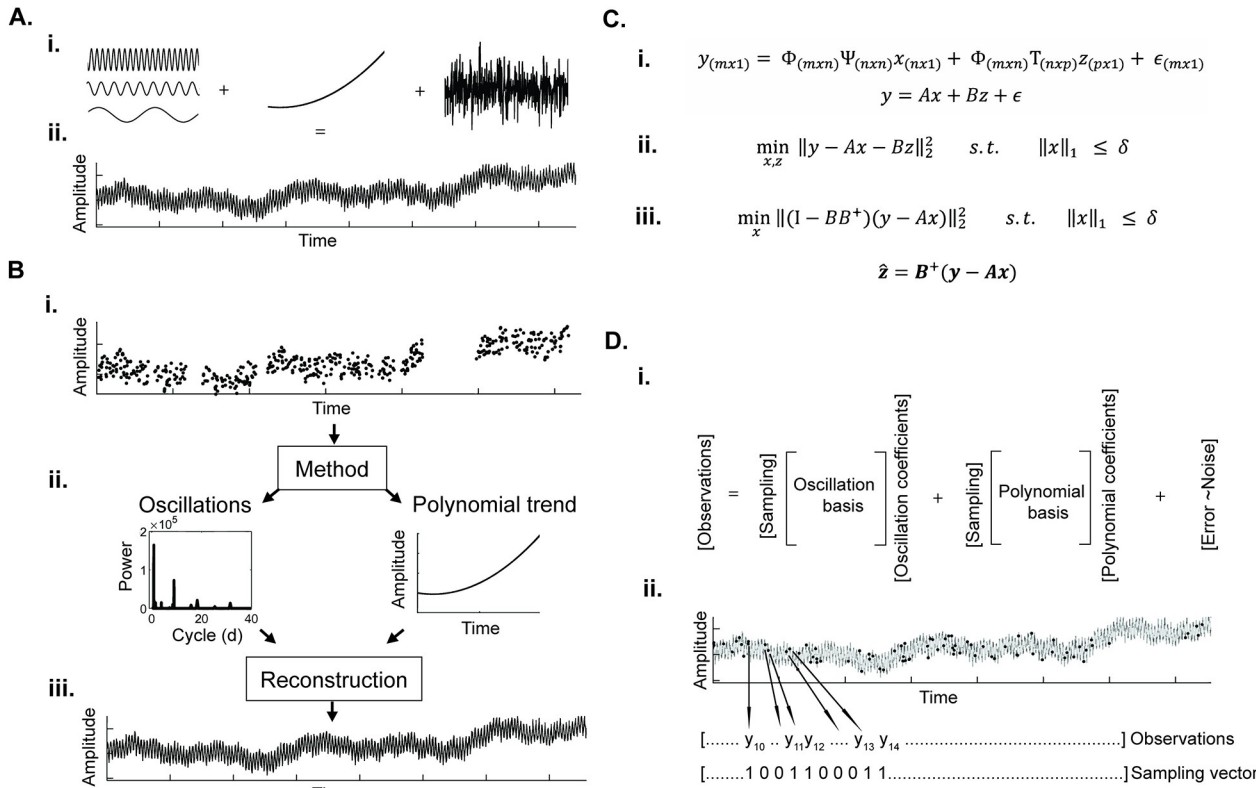

**Fig 2. Description of method objectives and signal assumptions.** (A) The core assumption of the model is that the underlying signal (ii) is the linear sum of (i) oscillations, a polynomial trend, and noise. Part (B) describes the overall workflow including (i) data input to the model, (ii) outputs, and (iii) estimated signal reconstructions. (Ci) Equation representing the core signal assumption that the observations come from a combination of oscillations, a polynomial trend, and noise or error. Notation includes $y$ (m x 1 vector of observed data), $\Psi$ (n x n discrete cosine transform (DCT) basis), $\Phi$ (m x n binary row subsampling matrix), $x$ (n x 1 DCT coefficients), T (n x p Vandermonde matrix), $z$ (p x 1 polynomial coefficients), (m x 1 error terms). (Cii) Expression for basis pursuit denoising containing $x$ and $z$ as unknowns, yielding a 2D minimization problem that is reduced to a 1D minimization problem (Ciii) by variable projection. (D) Schematic representation of the equations and sampling approach in (C).

human subjects by taking the densely sampled long-term, real-world timeseries, sparsely sampling it at different densities and paradigms, and applying BPWP.

## Cycle identification in simulated IED-rate timeseries

The simulated IED rate timeseries is shown in Fig 3A. The CWT spectrogram (Fig 3B) and average CWT spectrum (Fig 3F) reflect the increased signal power at periods of one day and one month. The results of the 10-fold 75/25 cross-validation (Fig 3C) for $\delta$ parameter selection capture a clear optimal $\delta$ value that minimizes mean square error (MSE) for each random sampling rate tested.

The BPWP power spectrum (Fig 3F) derived from 5 random samples per day (1,800 total samples) (Fig 3D) shows significant peaks aligned with the CWT spectrum (derived from >26,000 samples). The insert in Fig 3F shows significant BPWP peaks in relation to the noise floor (purple) which was determined by reshuffling the samples and recalculating BPWP. The amplitude of the noise floor is nearly an order of magnitude lower than the significant BPWP outputs. The signal reconstruction based on BPWP spectral and polynomial coefficient estimates (Fig 3E) closely captures both the multiday cycles and polynomial trend present in the simulated timeseries.

The reliability of cycle detection by BPWP depends on cycle length and on the variance and signal to noise ratio (SNR) of the underlying signal. We simulated oscillations with cycle lengths from 1 to 120 days with high and low variance and SNR conditions (S1 Supplementary Materials and S2AI and S2Bi Fig). Repeat sampling and recalculation of BPWP of these oscillations were used to determine the percent detection rate of the known oscillation in each condition. The reliability of cycle detection was worst in the low variance, low SNR, and lowest sampling rate conditions (S2 Fig). At a low sampling rate, a relationship between cycle length and detection rate becomes evident; detection rate decreases with increasing cycle period (S2 Fig).

## Cycle identification in real-world IED rate timeseries

The results of 10-fold 75/25 cross validation to select a $\delta$ parameter for each subject are shown in S1 Fig. The values of $\delta$ that yielded the lowest MSE for different sampling rates were consistent within subjects. Although the optimal $\delta$ value was in a similar range between subjects, we selected patient-specific $\delta$ values to minimize under or over-fitting by fixing $\delta$ across subjects.

The raw, hourly IED rate data from participant 1 show multiday cycles and a gradual increase in rate over time (Fig 4A). This is reflected in the CWT spectrogram (Fig 4B) of the IED timeseries which illustrates the stable circadian cycle and multiday cycles around two and four weeks. The average CWT spectrum across time is depicted in Fig 4F and demonstrates that this participant's IED rate component oscillations include periods around one, 18, 30, 50, and 100 days. Raw IED rate timeseries for the other participants are available in S3 and S4 Figs.

Fig 4D shows the 1800 random samples (5 per day) from the raw IED rate timeseries of participant 1. The sparse samples represent an over ten-fold decrease in sample density from the standard timeseries. The sparse data were input to BPWP to yield the narrowband DCT power spectrum (Fig 4F). The BPWP spectrum aligns visually with the main peaks in the CWT spectrum, capturing the highest amplitude circadian and multiday cycles. The "noise floor", BPWP spectral outputs from repeatedly (100x) reshuffling the original data in time (while preserving the original inter-sample intervals) and re-calculating BPWP, is plotted in purple. Stars denote the spectral peaks whose power exceeded the 99[th] percentile of the noise floor's power for that period. The power of the BPWP noise floor (Fig 4F) is an order of magnitude lower than the BPWP output. The BPWP spectrum and polynomial trend were used to create the signal

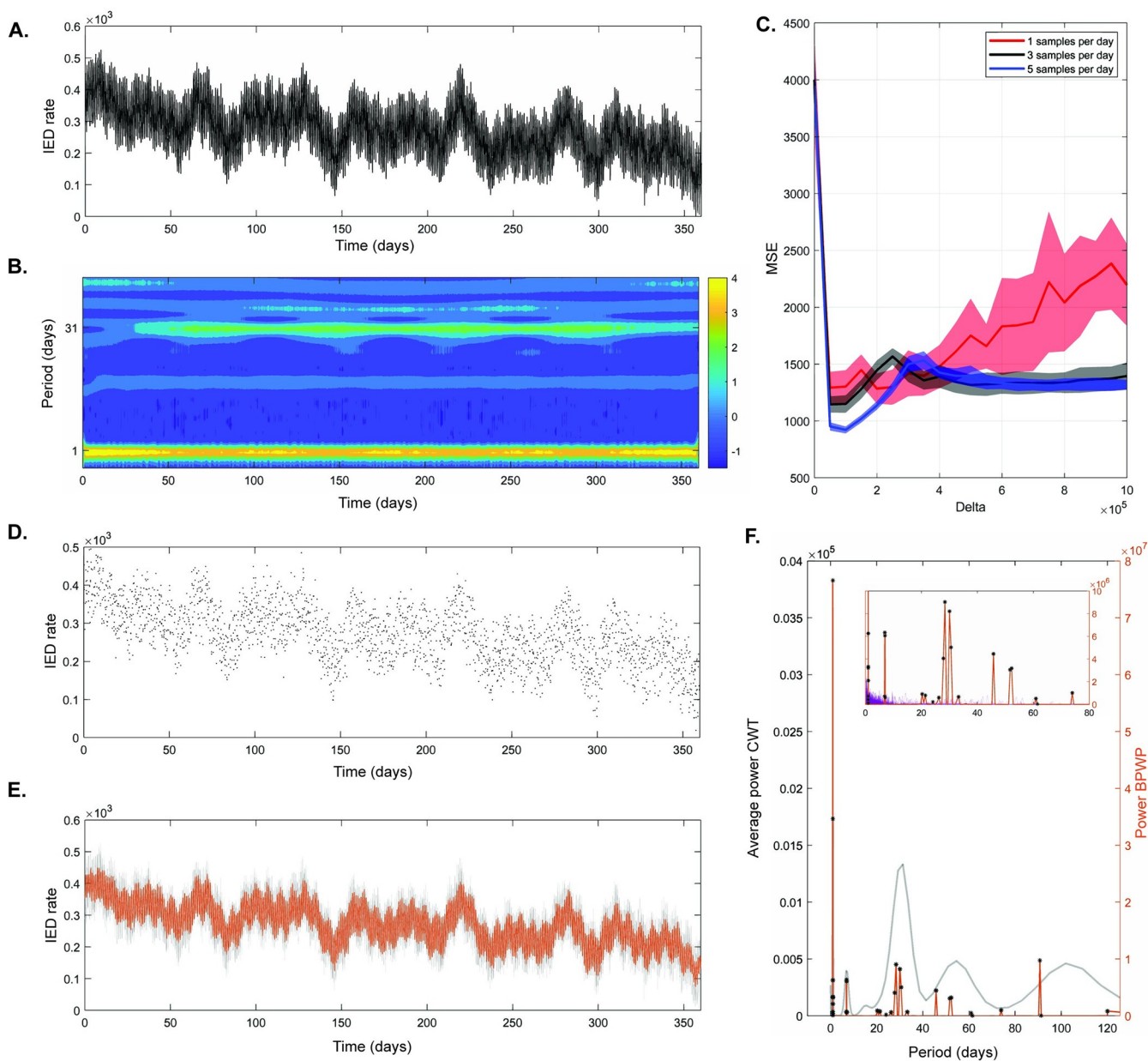

**Fig 3. Basis Pursuit with Polynomial Detrending (BPWP) outputs for simulated inter-ictal epileptiform discharge (IED) timeseries.** Raw data show hourly rate of IEDs, updated every 20 minutes, from a simulated signal containing oscillations at 1, 7, 15, 21, 30, 50, and 100 days. Timeseries consists of over 20,000 samples. (B) Complex wavelet transform (CWT) spectrogram of the timeseries in A showing power in different cycles. Color bar indicates spectral power. (C) Results of ten-fold 75/25 cross-validation for $\delta$ parameter selection. Average mean square error with 95% confidence intervals for each $\delta$ value and sampling rate tested are shown. (D) Random samples from the raw data in A averaging at five samples per day. Timeseries consists of 1,800 total samples. (E) Underlying raw data are shown in gray. The estimated reconstruction of the underlying data based on the sparse samples in (D) is shown in orange. (F) The average CWT spectrum (average over time from the spectrogram in (B) is shown in gray). The method's spectral output is shown in orange. Black stars denote significant peaks; peaks whose amplitude was above the 99[th] percentile of the distribution created by shuffling the input data and re-calculating the method 100 times. The insert shows the spectral outputs from the reshuffling in purple. The narrowband peaks from the method align with the central tendencies of the broadband CWT peaks.

reconstruction in Fig 4E. This estimated, underlying signal, closely aligns with the original raw data (Fig 4A) including multiday cycles and the gradual increase in average IED rate over time.

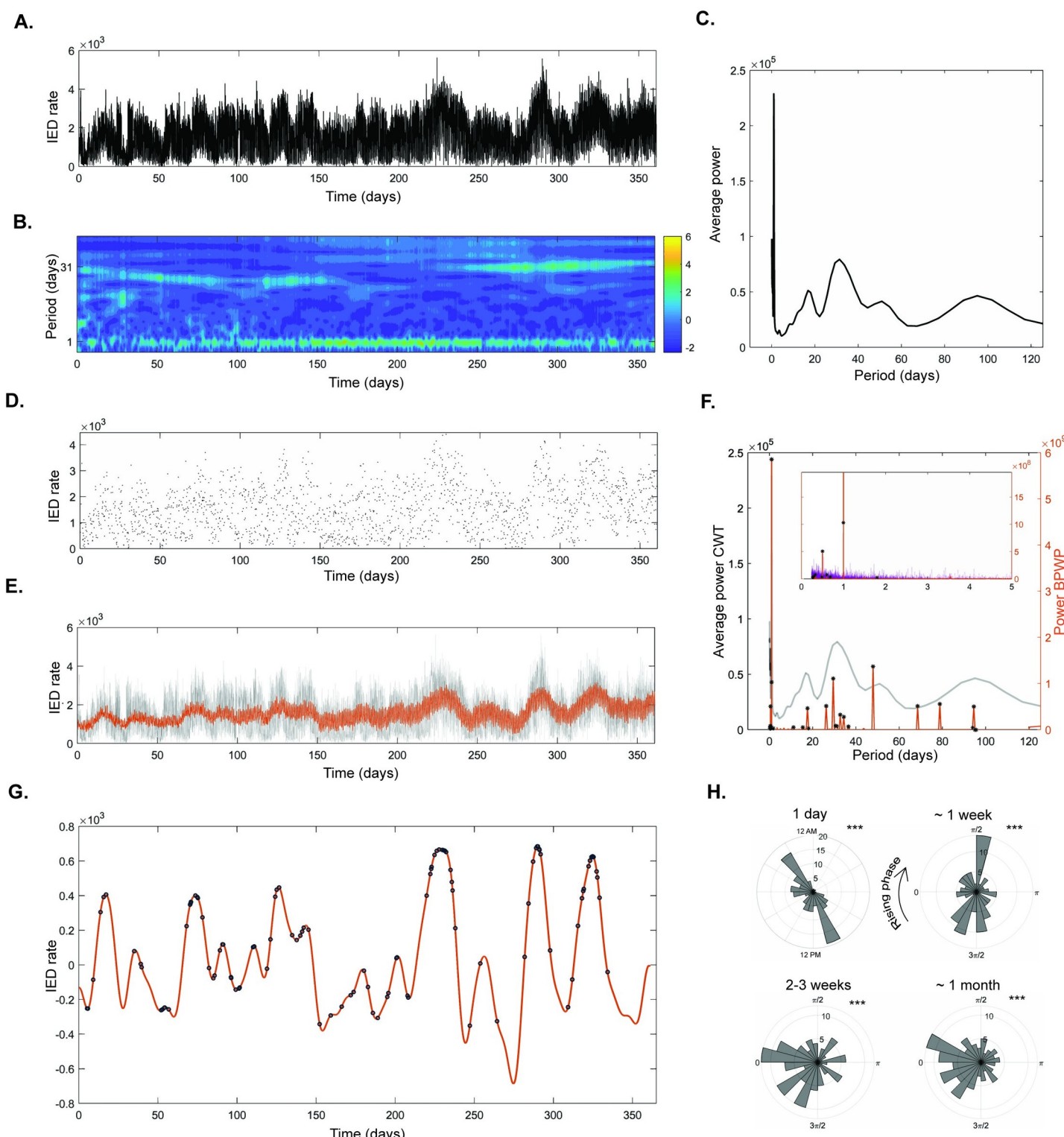

**Fig 4. Basis Pursuit with Polynomial Detrending (BPWP) of real-world inter-ictal epileptiform discharges (IED) timeseries, participant 1.** (A) Raw data showing hourly rate of IEDs detected from the left hippocampus, updated every 20 minutes. Timeseries consists of over 20,000 samples. (B) Complex wavelet transform (CWT) spectrogram of the timeseries in A showing power in different cycles. Strong cycles are evident at one day and around on month. (C) Average of CWT spectrum (averaged over time from the spectrogram in) (B) shows cycles of IED rate including periods around one day, two-three weeks, one month, fifty days, and 100 days. (D) Random samples from the raw data in A averaging at 5 samples per day. Timeseries consists of 1,800 total samples. (E) Underlying raw data are shown in gray. The method's estimated reconstruction of the underlying data based on the sparse samples in (D) is shown in orange. (F) The CWT spectrum for the raw data is shown in

gray. The method's spectral output is shown in orange. Black stars denote significant peaks; peaks whose amplitude was above the 99th percentile of the distribution created by shuffling the input data and re-calculating the method 100 times. The insert shows the spectral outputs from the reshuffling in purple. The amplitude of this noise floor is an order of magnitude smaller than the spectral output from the correctly ordered input data. The narrowband peaks from the method align with the central tendencies of the broadband CWT peaks. (G) The method's estimated reconstruction of the underlying signal using only significant peaks from F with a period longer than two days is in orange. Overlayed black circles denote when seizures occurred. Seizures appear to prefer the peaks of the combined slow cycles derived from the method. (H) The method-based reconstruction was filtered in cycle ranges around one day, one week, two to three weeks, and one month then the signal Hilbert transform was used to identify the phase at which seizures occurred for each of these cycles. Polar histograms denoting the phase at which seizures occurred for each of these cycles indicate a cycle-specific phase preference for seizures. Stars denote p < 0.001 on the Omnibus test for uniformity, indicating that seizure phase is not uniformly distributed.

Next, we determined if the reconstructed signal recapitulated the established relationship between cycles of IED rate and seizure occurrence [1]. To visualize the relationship between seizures and the multiday cycles identified by BPWP, we plotted a reconstruction consisting of only the oscillations that were 1) significant and 2) longer than two days (Fig 4G). For participant 1, the seizures occur near the peaks. Fig 4H shows polar histograms of the phase at which seizures occurred in multiday cycles that are strongly conserved between patients with focal epilepsy [2]. Seizures show cycle-specific phrase preferences (Fig 4H). The seizure-phase histograms for each cycle were significant on the Omnibus/Hodges-Ajne test [25] (p < 0.001) indicating non-uniform phase distribution and cycle-specific phase preferences. This indicates that the IED cycles identified by BPWP are pathophysiologically relevant.

BPWP outputs for the other two participants are available in S3 and S4 Figs. Overall BPWP performance is similar for participants 1 and 2 (Figs 4 and S3), both of which show multiday cycles evident in the raw data. Multiday cycles were less obvious in the raw data for participant 3 and similarly in the BPWP outputs (S4 Fig). Seizures in all three participants show a clear unimodal or bimodal phase preference within the circadian and multiday cycles.

### Real-world data loss: Impact of sampling density

Examples of different random sampling rates and associated BPWP outputs are shown in Fig 5. For all but the shortest CWT cycles (< 1 day), the scaled offset with the nearest BPWP peak decreases as the number of random samples per day increases, indicating that overall performance improves as more samples are available (Fig 5H). Offsets stabilize at around 3 samples per day and above. Sampling density results for participants 2 and 3 echo this relationship (S5 and S6 Figs).

### Real-world data loss: Impact of data drops

Contiguous data drops of 12-, 30-, and 60-days duration were created in the IED rate timeseries and the remaining data were sampled at a rate of 5 samples per day. BPWP spectral outputs under these conditions (Fig 6C, 6E, and 6G) varied in peak amplitude and position. BPWP reconstructions during the dropped windows had worse alignment with the original data than for the windows where data were available (Fig Bii, 6Dii, and 6Fii). The fidelity of the BPWP reconstructions was better in the case of shorter, more frequent drops (12 days) than for the longer, less frequent drops (60 days). The impact of data drops is shown for participants 2 and 3 in S7 and S8 Figs.

### Method performance: Frequency dependence

Given the model's use of sparse, irregular sampling on the order of a few samples per day, we anticipated that the overall method performance would be frequency-specific with slower cycles being more reliably estimated than higher frequency cycles. We therefore used a frequency-specific approach to evaluate the method performance. Briefly, we bandpass filtered

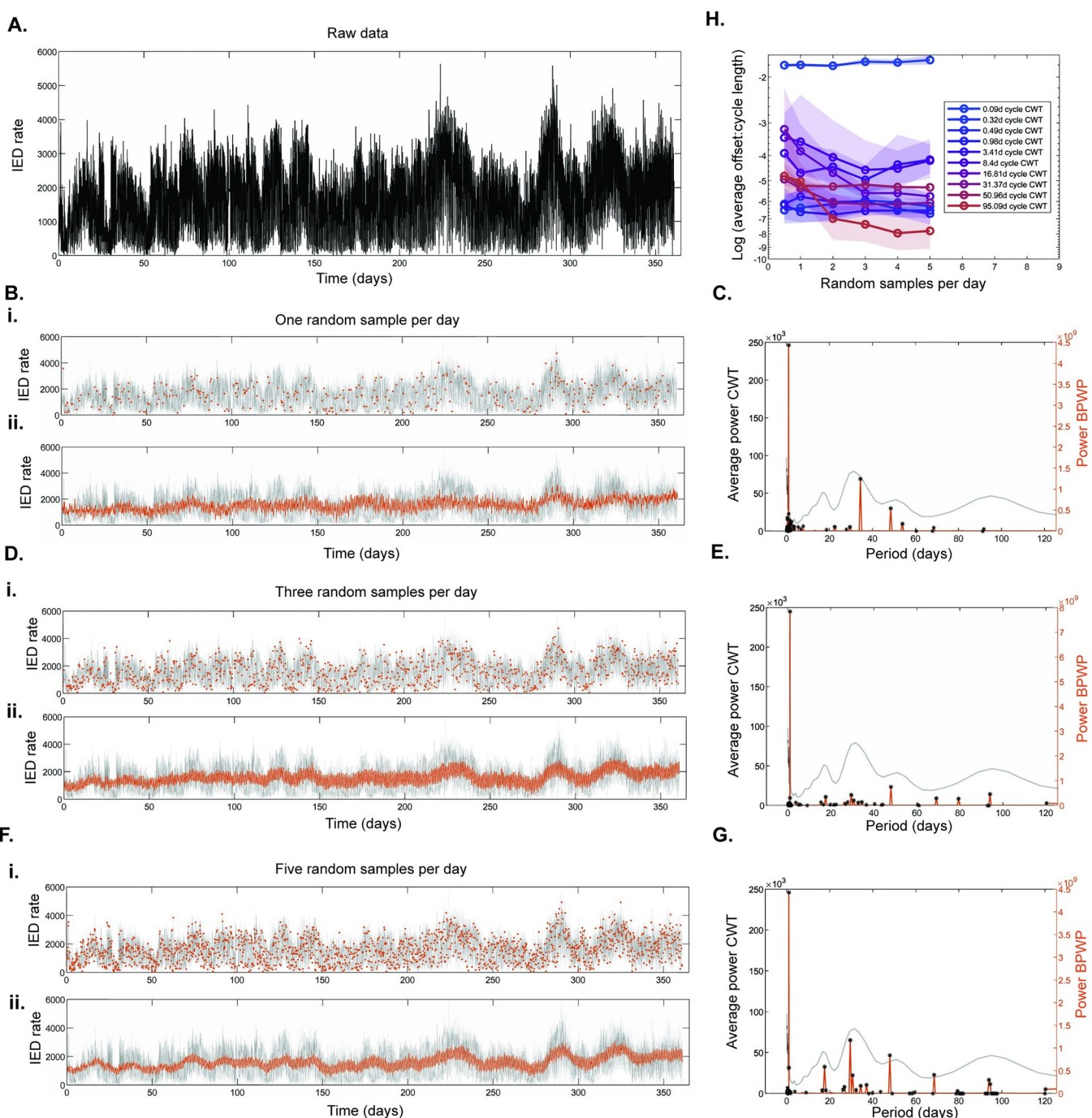

**Fig 5. Impact of varying sample density on Basis Pursuit with Polynomial Detrending (BPWP) outputs, participant 1.** (A) Raw data showing hourly rate of inter-ictal epileptiform discharges (IED) detected from the left hippocampus, updated every 20 minutes. Timeseries consists of over 20,000 samples. (Bi) Raw data are in gray and one random sample per day is in orange. (Bii) Raw data are in gray and the reconstructed signal using BPWP outputs based on input data of one random sample per day is in orange. (C) Average complex wavelet transform (CWT) spectrum from the raw data in (A) is in gray. The BPWP spectrum based on one sample per day input is shown in orange. Black stars denote significant peaks; peaks whose amplitude was above the 99[th] percentile of the distribution created by shuffling the input data and re-calculating the method 100 times. Random samples, signal reconstructions, and BPWP spectra are shown again for sampling rates of three and five per day in (D) and (E) and in (F) and (G) respectively. Agreement between BPWP output and raw data and CWT spectra improves as the signal is sampled more densely. Part (H) shows agreement between the BPWP and CWT spectra as a function of frequency of random sampling. For each sampling frequency, the raw data were resampled and BPWP was re-calculated 10 times. For each peak in the CWT spectrum, the offset between the period of the CWT peak and the nearest BPWP peak was calculated in terms of days and divided by the period of the CWT peak. This offset-to-cycle-length ratio was averaged across the 10 iterations and plotted as a log value on the y axis. The associated frequency of random sampling was plotted on the x axis. Shaded areas denote 95% confidence intervals. The offset ratio decreases and stabilizes as sampling density increases.

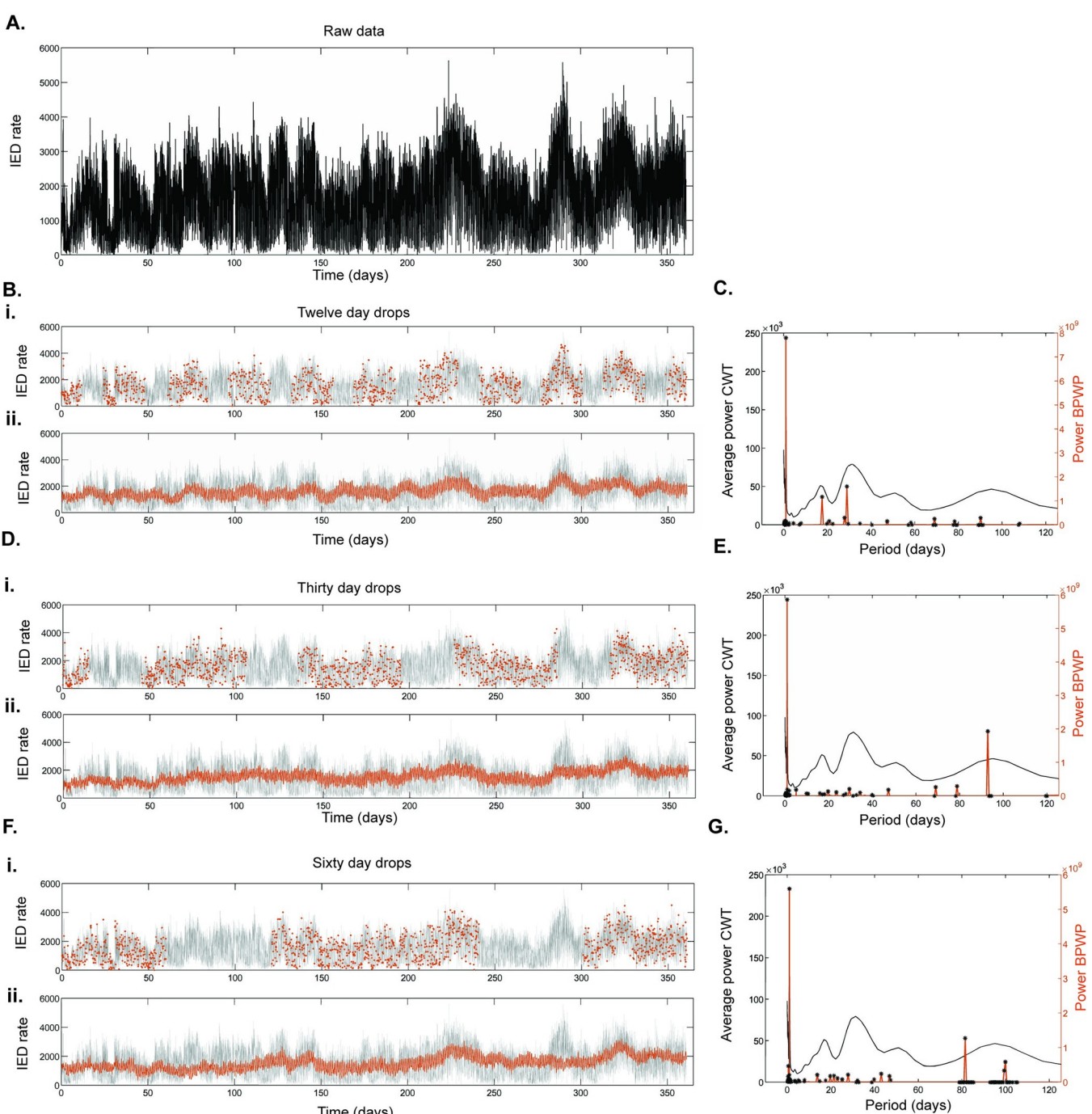

**Fig 6. Impact of data drops on Basis Pursuit with Polynomial Detrending (BPWP) outputs, participant 1.** (A) Raw data showing hourly rate of inter-ictal epileptiform discharges (IED) detected from the left hippocampus, updated every 20 minutes. Timeseries consists of over 20,000 samples. (Bi) Raw data are in gray and random sampling excluding 12-day data drops are in orange. (Bii) Raw data are in gray and the reconstructed signal using model output based on input data with 12-day data drops is in orange. (C) Average complex wavelet transform (CWT) spectrum from the raw data in (A) is in gray. The BPWP spectral output based on the sampling in Bi input is shown in orange. Black stars denote significant peaks; peaks whose amplitude was above the 99th percentile of the distribution created by shuffling the input data and re-calculating the method 100 times. Data drops of thirty- and sixty- days duration, signal reconstructions, and method spectra are shown in (D) and (E) and in (F) and (G) respectively. The total number of samples for BPWP is fixed across the conditions at n = 1307 which is approximately 4 samples per day assuming no drops.

both the original, raw, IED rate and the BPWP model output with central periods ranging from 1 to 120 days and compared the two with the Pearson correlation coefficient (Fig 7A). For participant 1, correlation coefficients were lowest (around 0.5) for short cycles below 20 days and highest (approaching 1) for the longer cycles above 30 days (Fig 7B). Correlation coefficients were significant for each cycle evaluated with p < 0.001. Examples of the filtered timeseries for cycles around 1, 19, and 100 days are shown in Fig 7C. At the higher frequencies with lower correlation coefficients, the model output tends to be lower amplitude and intermittently out of phase with the original data. At the lower frequencies where the correlation coefficients were highest, the two timeseries are very closely aligned with similar amplitude. Although, among faster cycles, correlation was highest at one day, then dipped between one and twenty-days. This may reflect phase instability in cycles at this range, coupled with the comparatively strong circadian cycle. Participants 2 and 3 showed similar relationships between cycle length and strength of correlation (S10 and S11 Figs).

## Discussion

BPWP successfully identified multiday cycles of real-world IED rates from human epileptic hippocampus using a sparsely sampled fraction of the samples required of traditional approaches for frequency decomposition. We anticipate BPWP will enable frequency decomposition of sparse neuro-behavioral timeseries in a variety of applications. BPWP may guide efficient sampling of neural features on implanted devices, extending storage and minimizing patient burden.

Basis pursuit denoising (BPDN) [21,22], like the similar and at times identical formulation of LASSO [23], leverages the properties of the L1 norm. L1 regularization promotes sparsity and robustness to outliers and conducts built-in feature selection. The degrees of freedom is straightforward to derive as well, and is unbiasedly estimated as the number of nonzero coefficients [26,27]. Assuming the key assumptions (sparsity and separability) are met, these properties lend themselves to reliable model outputs.

We modified BPDN to include a polynomial representation that manages non-stationarity due to slow changes in signal mean over time. Although low-order polynomial bases have been directly incorporated into sparse recovery approaches before [28,29], the typical approach to polynomial trends in neural signal processing is to remove them during preprocessing. Polynomial detrending as a preprocessing step is highly debated, as the choice of technique can have a non-negligible influence on the slow fluctuations in the data [30]. By directly including polynomial fitting in the model, we nullify polynomial detrending as a preprocessing step and appropriately consider trends and oscillations in parallel. Non-stationarity in the frequency domain, such as oscillation phase shifts or the emergence and disappearance of oscillations are not explicitly addressed by this model. Our implementation assumes stationarity in the frequency domain. Future directions include better accommodating these spectral non-stationarities.

BPWP method performance is likely influenced by several factors including the frequency of the oscillations being reconstructed, random sampling rate, and stationarity of the underlying process. We found that the correlation between the estimated model outputs and the original data were frequency-dependent with strongest correlations approaching a value of 1 for the one day, and thirty day and longer cycles. These findings likely reflect the increased uncertainty in estimating faster cycles with as few as 5 random samples per day. Although the model output and original data were significantly correlated at these faster cycles, the model output was intermittently out of phase with the original data at these frequencies. This may be the result of non-stationarity in the original data compounded by the low sampling rate. Overall,

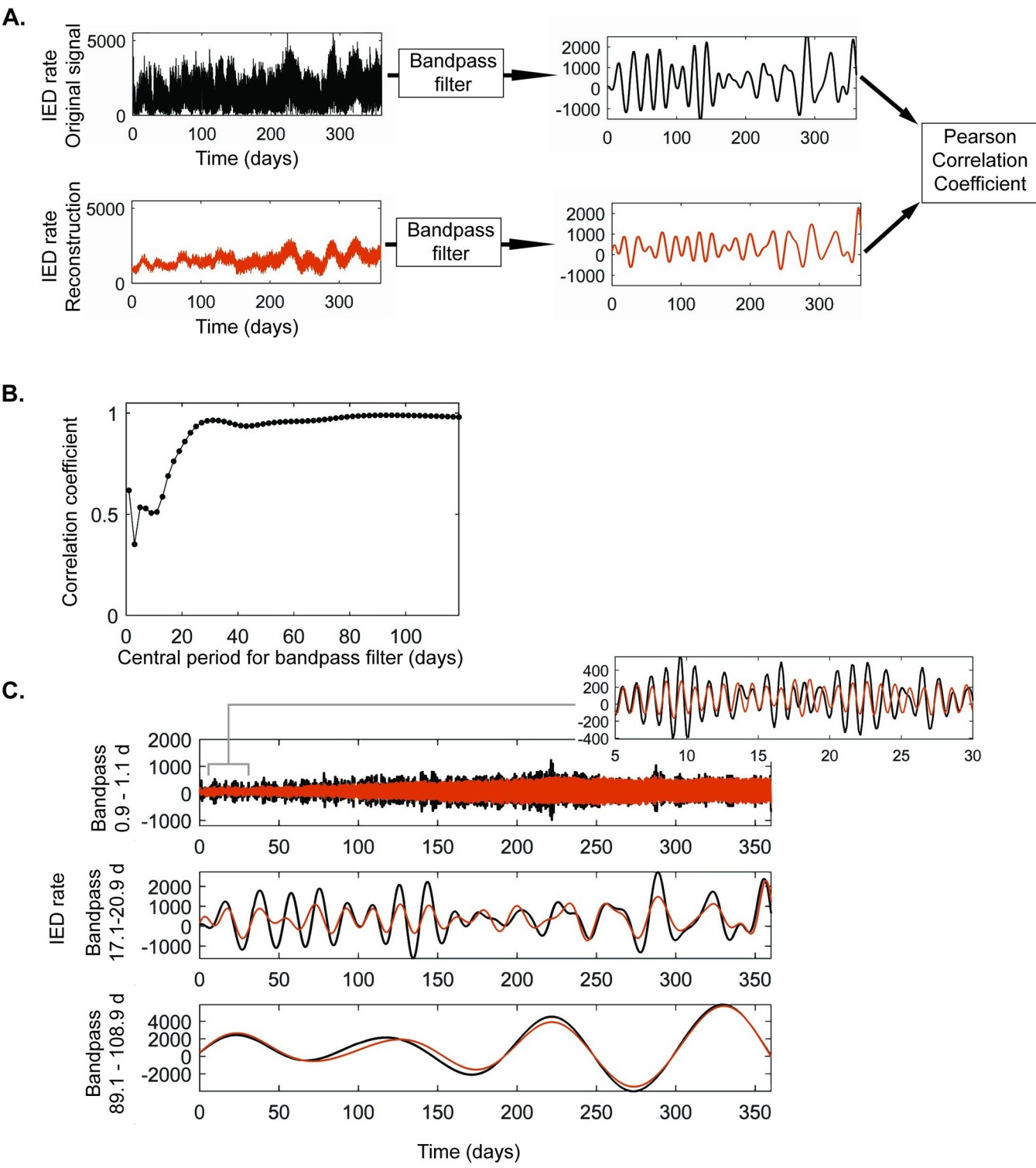

**Fig 7. Basis Pursuit with Polynomial Detrending (BPWP) method performance, participant 1.** (A) Diagram depicting the approach to comparing the original inter-ictal epileptiform discharge (IED) rate timeseries (black) with the BPWP model output (orange). Each timeseries was bandpass filtered and the two filtered timeseries were used to calculate the Pearson correlation coefficient. (B) The Y axis shows the Pearson correlation coefficient for each bandpass filter with central periods listed on the X axis. P value for each point depicted was < 0.001. (C) Examples of filter outputs for the approximate daily (top), 19-day (middle), and 100 day (bottom) cycles. The original signal is in black, the BPWP output is in orange. The insert describes the daily cycle filter outputs from the 5 to 30-day time points.

BPWP performance was strong considering the relative paucity of datapoints input to the model.

There are numerous alternatives to BPDN to detect periodic signals in irregularly sampled data. For example, the Lomb-Scargle periodogram is used within astronomy to identify periodicities in unevenly sampled data [31–33]. Lomb-Scargle and other approaches leverage Fourier transforms and least squares or chi-squared fitting [34], which, compared to L1-regularization can yield less-sparse solutions and increase susceptibility to outliers [31,32]. Other approaches use continuous-time autoregressive moving average models and Bayesian probability theory [35]. Additionally, not all Fourier-based approaches are bound by regular sampling as is the Discrete Fourier Transform (DFT) [16]. Non-Uniform Discrete Fourier Transform can accommodate irregular sampling [36] with the implementation by Fessler and colleagues being among the most common given its computational efficiency and generalization to multidimensional signals [37]. We chose to work from the BPDN framework because it is a convex method and includes L1 regularization which enables conservative estimates of component cycles. Though, we acknowledge that alternatives exist and may be more appropriate for different applications.

The computational efficiency of BPWP is limited by several factors. Large NxN matrices can slow computation times, though their exact dimensions depend on the application and number of coefficients being evaluated. The most time-consuming step of this approach is the preliminary, patient-specific parameter sweep by which the $\delta$ parameter that constrains the L1 norm is defined. Depending on the machine, parallel processing capabilities, density of the parameter sweep, number of iterations used to build the non-parametric reference distribution, and N, identifying $\delta$ can take a day to days. Though, given ongoing rapid advancements in processing capacity, we do not consider this a serious limitation. Nonetheless, the workflow presented here is not currently suitable for real-time computations on clinically implanted devices.

Real-world data loss had differential impacts on the reliability of the BPWP outputs. BPWP spectral outputs were reasonably stable above a sampling frequency of approximately 3 random samples per day for the IED rate data, and remarkably recovered the known underlying circadian and longer timescale cycles in the data. Long data drops on the order of months resulted in variable spectra and inaccurate reconstructions in unsampled epochs. BPWP may be more suitable for circumstances of infrequent but consistent sampling, than for situations with long periods of data loss.

The suitability of BPWP for additional neuro-behavioral applications depends in part on the underlying signal. In simulation, the reliability of cycle detection depended on SNR, sampling rate, and variance. In applications where the ground truth is unavailable, hypothesis driven experiments and cautious interpretation of the model output are required. For example, behavioral timeseries such as Likert scaled ecological momentary assessments from participants who tend to use only a narrow range of the scale (low variance and SNR) are unlikely to reveal cycles. Additionally, the sampling paradigm is an important consideration for retrospective application of BPWP to sparse neuro-behavioral timeseries. BPWP requires irregular samples in time because samples collected at regular intervals, such as a specific time of day, introduce false cycles at the interval duration.

We extrapolated from raw LFP to IED rates derived from LFP, showing that a secondary neuro-behavioral feature can be evaluated with sparse recovery. Comparable sparse recovery approaches in neural signal processing have primarily been applied to signal compression (including LFP), action potentials, and signal transmission [38,39], as opposed to higher order features such as IED rates. Although sparse recovery systems have been explored for data compression in implantable neural devices [40], our findings support the implementation of

feature sampling and storage paradigms with downstream basis pursuit-based reconstruction techniques in mind. Random and sparse sampling of epileptiform features could extend storage capacity in clinically approved devices and enable reliable comparison of electrophysiologically based features with behavioral and multi-domain data.

## Materials and methods

### Ethics statement

This research was conducted as part of a study evaluating the safety and feasibility of brain state tracking and modulation in temporal lobe epilepsy (ClinicalTrials.gov identifier: NCT03946618). This study was approved by the Mayo Clinic Institutional Review Board and the Food and Drug Administration (IDE G180224). Formal written consent was obtained from all participants.

### Application

The purpose of BPWP is to identify oscillations and polynomial trends in sparsely and irregularly sampled neuro-behavioral data. The real-world test data used here are IED rate timeseries. IEDs are defined here as sharp transients detected on LFP recordings from depth electrodes implanted in epileptogenic brain regions (Fig 1Bi) [41]. The IED detector from Janca and colleagues [41] is an adaptive, automated spike detector that models the statistical distributions of signal envelopes containing both the IEDs and background signal using a maximal likelihood estimation algorithm.

### Signal

We assume the underlying signal consists of oscillations, a polynomial trend, and gaussian noise (Fig 2A).

### Approach

The overall approach of BPWP is outlined in Fig 2B. Sparse samples are input to the method which estimates the oscillation coefficients and polynomial coefficients that describe the data. These outputs can then be used to estimate the continuous underlying signal.

### Model

The signal model is described in Eqs 1 and 2 and Fig 2C,

$$y_{(mx1)} = \Phi_{(mxn)}\Psi_{(nxn)}x_{(nx1)} + \Phi_{(mxn)}T_{(nxp)}z_{(px1)} + \epsilon_{(mx1)} \tag{1}$$

$$y = Ax + Bz + \epsilon \tag{2}$$

where $y$ is a vector of samples of length $m$, $\Phi$ is the $m$ x $n$ binary row subsampling matrix, $\Psi$ is the $n$ x $n$ discrete cosine transform (DCT) basis, $x$ is the length $n$ vector of DCT coefficients, T is the $n$ x p Vandermonde polynomial basis, $z$ is the length $p$ vector of polynomial coefficients where $p$ is equal to the maximum polynomial degree under consideration plus one, and $\epsilon$ is the length m vector of error terms. $A$ is the product of $\Phi$ and $\Psi$. $B$ is the product of $\Phi$ and T. Schematic representation of the basis subsampling is depicted in Fig 2Dii.

The oscillation basis is a matrix of DCT-II coefficients and determines the frequencies of the component oscillations that can be resolved. The DCT-II expression is shown in Eq 3 [42]. In Eq 3, $s(n)$ is the signal at point $n$, $N$ is the total length of the signal $s$, and $\delta_{k1}$ is the Kronecker

delta.

$$\Psi_k = \sqrt{\frac{2}{N}} \sum_{n=1}^{N} s(n) \frac{1}{\sqrt{1 + \delta_{k1}}} \cos(\frac{\pi}{2N}(2n - 1)(k - 1))$$

$$k = 0 : \ N - 1 \tag{3}$$

The default frequency representation in the DCT is defined by the sampling rate and the number of points. The associated frequency vector has sparse representation at low frequencies, and dense representation at high frequencies. To ensure that the DCT basis encompasses an adequate density of low frequency coefficients, such as cycles on the order of months or days in length, we incorporated a frequency selection step to the construction of the DCT basis $\Psi$ (Eq 4). As in variable density sampling[15], the function $f(k)$ can be adapted to optimize the range, density, and central tendency of frequencies evaluated by a transform.

$$\Psi_{f(k)} = \sqrt{\frac{2}{N}} \sum_{n=1}^{N} s(n) \frac{1}{\sqrt{1 + \delta_{k1}}} \cos(\frac{\pi}{2N}(2n - 1)(f(k) - 1))$$

$$k = 0 : \ N - 1 \tag{4}$$

To minimize the disruption of the orthonormal basis structure, we made targeted modifications to the default frequency representation for the DCT via $f(k)$. Briefly, we increased the density of frequency representation within the range of interest on the low frequency end at the expense of randomly removing an equivalent number of frequencies from the high frequency end (S1 Supplementary Materials).

## Estimation

BPWP is based on BPDN (Eq 5) [21–23]. The parameter $\delta$ captures error and constrains the L1 norm of the DCT coefficient vector.

$$\min_{x} \ \|y - Ax\|_2^2 \quad s.t. \|x\|_1 \leq \delta \tag{5}$$

The polynomial representation is incorporated into BPDN in Eq 6. This expression has two unknowns, $x$ and $z$.

$$\min_{x,z} \ \|y - Ax - Bz\|_2^2 \quad s.t. \|x\|_1 \leq \delta \tag{6}$$

Using variable projection, we reduced the two-dimensional optimization problem to a one-dimensional minimization of $x$ (Eq 7), then use $x$ to solve for $z$ (Eq 8) [43, 44]. The derivation is available in S1 Supplementary Materials.

$$\min_{x} \ \|(I - BB^+)(y - Ax)\|_2^2 \quad s.t. \|x\|_1 \leq \delta \tag{7}$$

$$\hat{z} = B^+(y - Ax) \tag{8}$$

Eqs 7 and 8 constitute the core expressions of BPWP. Key assumptions of BPWP are discussed in detail in S1 Supplementary Materials.

The DCT and polynomial coefficient outputs can be used to reconstruct the underlying timeseries. Expressions for the signal reconstruction are available in S1 Supplementary Materials.

### Features and parameters

Several model components can be adjusted depending on the application. Special considerations are discussed in more detail in S1 Supplementary Materials.

### Solver

We used the SDPT3 solver [45,46] in CVX: MATLAB Software for Disciplined Convex Programming (CVX Research Inc.), a system designed for convex optimization in MATLAB (MathWorks) [47].

### Parameter sweeps

There are many model fits that minimize Eq 7. We used patient-specific, 10-fold 75/25 cross validation to identify an appropriate value for $\delta$ (S1 Supplementary Materials). By choosing $\delta$ based on cross-validation, we select a model that balances over- and under-fitting (the value of $\delta$ that minimizes mean square error for each participant).

### Determining significance

We repeatedly calculated DCT coefficients $x$ by reshuffling the observed samples, $y$, in time (randomly changing y value order with a fixed sampling matrix: preserving the original inter-sample intervals) 100 times to build a nonparametric distribution. Coefficients calculated from the original data were deemed significant if their amplitude was equal to or greater than the 99th percentile of the reshuffling-derived distribution. Examples of significant coefficients for nonparametric distributions built from 1,000 and 10,000 iterations are available for comparison in S12 and S13 Figs. We felt the spectra were sufficiently similar by visual inspection to carry the 100-iteration approach forward.

### Simulated IED rate timeseries

Simulated data with duration of 12 months and cycle durations comparable to those observed in long-term IED rates [1] were generated. Each sample reflected one hour of simulated IED counts, updated every 20 minutes. Component oscillations included 1-, 7-, 15-, 21-, 30-, 50-, and 100-day periods. Simulated oscillations were present throughout (stationary) the year-long signal duration. A slow trend was included as a first order polynomial.

### Participants

Three, adult, female patients with drug resistant temporal lobe epilepsy were implanted with the investigational Medtronic Summit RC+S device [8,48] as part of an investigational device exemption clinical trial (FDA IDE: G180224, https://clinicaltrials.gov/ct2/show/NCT03946618.) Patients provided written and informed consent in accordance with Institutional Review Board and Food and Drug Administration Requirements.

### IED rate timeseries

One year of continuous IED rate recordings was used from each of the three participants. IEDs were detected from left (the more epileptogenic hemisphere in all participants) hippocampal LFP recordings and logged in one-hour totals updated every 20 minutes [41]. IED rates were collected nearly continuously, with daily hour-long data drops for charging and occasional longer drops due to connectivity issues. For use in the continuous analysis, gaps in the timeseries were interpolated as previously described [1]. The peri-ictal period (two hours

prior to seizure, during seizures, and two hours following seizures) was omitted and interpolated as above. For the sparse analysis, the timeseries were randomly sampled according to the relevant paradigm.

### Comparison with complex wavelet transform

Complex wavelet transform (CWT) using Morlet wavelets of the real, original, densely sampled, timeseries was compared with the BPWP spectral outputs (MATLAB function cwt, Morlet wavelet, symmetry parameter = 3, time-bandwidth product = 60). The CWT was calculated as described previously [48]. To compare the broadband CWT and narrowband BPWP spectra across sampling conditions, we identified each peak in the CWT spectrum and calculated the offset in terms of period between the CWT peak and the nearest significant BPWP peak detected. The offset was then scaled by dividing it over the length of the CWT cycle. This was repeated for 10 different re-samplings under each condition and averaged.

### Simulating data drops in IED rate timeseries

Contiguous blocks of varying duration (12, 30, and 60 days; 120 total days dropped in each case) were omitted from random sampling to test BPWP performance with data drops. The random daily sampling rate was fixed for the remaining days.

### Seizure phase analysis

Seizures were detected from the raw, continuous LFP recordings [8,49]. Seizures were defined here as the events whereby seizure detector probabilities exceeded 0.99. To evaluate the phase of seizures within the BPWP reconstruction, the BPWP reconstruction was filtered with a least-squares finite impulse response (FIR) bandpass filter of order (3 * Nyquist frequency) with zero phase shift in filter in cycle ranges that are highly conserved in focal epilepsies [2,48].: 1, 5–10, 15–25, and 25–35 days. Instantaneous phase was calculated via the Hilbert transform. For each cycle, the phase at which seizures occurred was noted.

### Statistical analysis

The Omnibus (Hodges-Ajne) test for non-uniformity of circular data was used to determine if there was a phase preference for seizures for each multiday cycle evaluated [25]. Evaluating overall method performance by comparing the sparse reconstruction with the original raw IED rate timeseries presented a statistical a challenge as this required comparing a very sparse, low noise timeseries with a densely sampled and comparatively higher noise timeseries. We also anticipated that model performance would be cycle specific. The infrequent, irregular sampling rate (5 samples per day) may result in poorer performance at high frequencies such as the daily cycle, but adequate performance at lower frequencies such as the 60-day cycle. Therefore, we chose a frequency-specific approach to explore the performance of the BPWP method. We filtered the original raw signal and the reconstructed timeseries using the same bandpass FIR filter described above in the seizure phase analysis. Central periods from 1 day to 120 days with a lower limit of 90% of the central period and upper limit of 110% of the central period defined each band. After the original raw signal was filtered, it was down sampled in time to match the reconstruction's time vector, ensuring pointwise comparisons between the original data and reconstruction would be time locked. These two filtered timeseries were then compared with the Pearson correlation coefficient (MATLAB function: corrcoef) with a significance level of 0.05.

## Code and processing

Analyses were run on a computer with Intel Xeon Silver 4108 CPU @ 1.80 GHz, 188 GB RAM, 16 physical cores, and 32 logical cores, and running Ubuntu version 18.04.6. Runtimes in our system were largely dictated by the number of iterations by which the nonparametric reference distributions were built. This scaled linearly. Using participant 1 as an example, the 100, 1,000, and 10,000 iteration cases took a few hours, 1.5 days, and 12.7 days. Code, deidentified raw data, and simulated data are available on GitHub at (https://github.com/irenabalzekas/BPWP).

## Supporting information

**S1 Supplementary Materials. Supplementary methods.**
(DOCX)

**S1 Fig. Subject-specific 10-fold 75/25 cross-validation for delta parameter selection.** Mean square error (MSE) between the real and estimated samples was calculated for each iteration of $\delta$ parameter and density of random sampling. (i) Shows results from the simulated IED time-series. Subplots (ii), (iii), (iv) show results from real-world IED timeseries from participants 3, 1, and 2 respectively.
(TIF)

**S2 Fig. Impact of variance, SNR, and cycle length on cycle detection in silico.** Signals of different cycle length, variance, and SNR were generated and input to BPWP. (A) Low variance condition. (i) Example signals with low and high SNR are shown in blue and red. Y axis labels with "d. c." denotes "day cycle", capturing the period for each simulated oscillation. (ii) Simulated signals were resampled and BPWP was recalculated 10 times. The percent of iterations wherein the true oscillation was captured as a significant peak constitutes the percent detected metric on the y axis. Detection rate is plotted for a low sampling (1 random sample per week) and higher sampling (4 random samples per week) condition. Part (B) mirrors part (A), but for simulated signals in the high variance condition.
(TIF)

**S3 Fig. BPWP outputs for real world IED timeseries, participant 2.** (A) Raw data showing hourly rate of IEDs detected from the left hippocampus, updated every 20 minutes. Timeseries consists of over 20,000 samples. (B) Complex wavelet transform (CWT) spectrogram of the timeseries in A showing power in different cycles. Strong cycles are evident at one day and around on month. (C) Random samples from the raw data in A averaging at 5 samples per day. Timeseries consists of 1,800 total samples. (D) Underlying raw data are shown in gray. BPWP's estimated reconstruction of the underlying data based on the sparse samples in (C) is shown in orange. (E) The method's estimated reconstruction of the underlying signal using only significant peaks from (G) with a period longer than two days is in orange. Overlayed black circles denote when seizures occurred. Seizures appear to prefer the peaks of the combined slow cycles derived from the method. (F) Average of CWT spectrum (averaged over time from the spectrogram in (B) shows cycles of IED rate including periods around one day, two-three weeks, one month, and 90 days. (G) The CWT spectrum for the raw data is shown in gray. The method's spectral output is shown in orange. Black stars denote significant peaks; peaks whose amplitude was above the 99[th] percentile of the distribution created by shuffling the input data and re-calculating the method 100 times. The insert shows the spectral outputs from the reshuffling in purple. (H) The BPWP-based reconstruction was filtered in cycle ranges around one day, one week, two to three weeks, and one month then Hilbert

transformed to identify the phase at which seizures occurred for each of these cycles. Polar histograms denoting the phase at which seizures occurred for each of these cycles indicate a cycle-specific phase preference for seizures. Stars denote p < 0.001 on the Omnibus test for uniformity, indicating that seizure phase is not uniformly distributed.
(TIF)

**S4 Fig. BPWP outputs for real world IED timeseries, participant 3.** (A) Raw data showing hourly rate of IEDs detected from the left hippocampus, updated every 20 minutes. Timeseries consists of over 20,000 samples. (B) Complex wavelet transform (CWT) spectrogram of the timeseries in A showing power in different cycles. Strong cycles are evident at one day and around on month. (C) Random samples from the raw data in (A) averaging at 5 samples per day. Timeseries consists of 1,800 total samples. (D) Underlying raw data are shown in gray. The estimated reconstruction of the underlying data based on the sparse samples in (C) is shown in orange. (E) The method's estimated reconstruction of the underlying signal using only significant peaks from F with a period longer than two days is in orange. Overlayed black circles denote when seizures occurred. (F) Average of CWT spectrum (averaged over time from the spectrogram in B) shows cycles of IED rate including periods around one day, two weeks, and one month. (G) The CWT spectrum for the raw data is shown in gray. The method's spectral output is shown in orange. Black stars denote significant peaks; peaks whose amplitude was above the 99th percentile of the distribution created by shuffling the input data and re-calculating the method 100 times. The insert shows the spectral outputs from the reshuffling in purple. (H) The method-based reconstruction was filtered in cycle ranges around one day, one week, two to three weeks, and one month then Hilbert transformed to identify the phase at which seizures occurred for each of these cycles. Polar histograms denoting the phase at which seizures occurred for each of these cycles indicate a cycle-specific phase preference for seizures. Stars denote p < 0.001 on the Omnibus test for uniformity, indicating that seizure phase is not uniformly distributed.
(TIF)

**S5 Fig. Impact of varying sample density on BPWP outputs, participant 2.** (A) Raw data showing hourly rate of IEDs detected from the left hippocampus, updated every 20 minutes. Timeseries consists of over 20,000 samples. (Bi) Raw data are in gray and one random sample per day are in orange. (Bii) Raw data are in gray and the reconstructed signal using model outputs based on input data of one random sample per day is in orange. (C) Average complex wavelet transform (CWT) spectrum from the raw data in (A) is in gray. BPWP spectral output based on one sample per day input is shown in orange. Black stars denote significant peaks; peaks whose amplitude was above the 99th percentile of the distribution created by shuffling the input data and re-calculating BPWP 100 times. Random samples, signal reconstructions, and BPWP spectra are shown again for sampling rates of three and five per day in (D) and (E) and in (F) and (G) respectively. Agreement between BPWP output and raw data and CWT spectra improves as the signal is sampled more densely. Part (H) shows agreement between the BPWP and CWT spectra as a function of frequency of random sampling. For each sampling frequency, the raw data were re-sampled and BPWP was re calculated 10 times. For each peak in the CWT spectrum, the offset between the period of the CWT peak and the nearest BPWP peak was calculated in terms of days and divided by the period of the CWT peak. This offset-to-cycle-length ratio was averaged across the 10 iterations and plotted as a log value on the y axis. The associated frequency of random sampling was plotted on the x axis. Shaded areas denote 95% confidence intervals. The offset ratio decreases and stabilizes as sampling density increases.
(TIF)

**S6 Fig. Impact of varying sample density on BPWP outputs, participant 3.** (A) Raw data showing hourly rate of IEDs detected from the left hippocampus, updated every 20 minutes. Timeseries consists of over 20,000 samples. (Bi) Raw data are in gray and one random sample per day are in orange. (Bii) Raw data are in gray and the reconstructed signal using model outputs based on input data of one random sample per day is in orange. (C) Average complex wavelet transform (CWT) spectrum from the raw data in (A) is in gray. The BPWP spectral output based on one sample per day input is shown in orange. Black stars denote significant peaks; peaks whose amplitude was above the 99[th] percentile of the distribution created by shuffling the input data and re-calculating BPWP 100 times. Random samples, signal reconstructions, and BPWP spectra are shown again for sampling rates of three and five per day in (D) and (E) and in (F) and (G) respectively. Agreement between BPWP output and raw data and CWT spectra improves as the signal is sampled more densely. Part (H) shows agreement between the BPWP and CWT spectra as a function of frequency of random sampling. For each sampling frequency, the raw data were resampled and the BPWP was re calculated 10 times. For each peak in the CWT spectrum, the offset between the period of the CWT peak and the nearest BPWP peak was calculated in terms of days and divided by the period of the CWT peak. This offset-to-cycle-length ratio was averaged across the 10 iterations and plotted as a log value on the y axis. The associated frequency of random sampling was plotted on the x axis. Shaded areas denote 95% confidence intervals.
(TIF)

**S7 Fig. Impact of data drops on BPWP outputs, participant 2.** (A) Raw data showing hourly rate of IEDs detected from the left hippocampus, updated every 20 minutes. Timeseries consists of over 20,000 samples. (Bi) Raw data are in gray and random sampling excluding 12-day data drops are in orange. (Bii) Raw data are in gray and the reconstructed signal using model output based on input data with 12-day data drops is in orange. C) Average complex wavelet transform (CWT) spectrum from the raw data in (A) is in gray. The BPWP spectral output based on the sampling in (Bi) input is shown in orange. Black stars denote significant peaks; peaks whose amplitude was above the 99th percentile of the distribution created by shuffling the input data and re-calculating BPWP 100 times. Data drops of thirty- and sixty- days duration, signal reconstructions, and method spectra are shown in (D) and (E) and in (F) and (G) respectively. The total number of samples for BPWP is fixed across the conditions at n = 1307 which is approximately 4 samples per day assuming no drops.
(TIF)

**S8 Fig. Impact of data drops on BPWP outputs, participant 3.** (A) Raw data showing hourly rate of IEDs detected from the left hippocampus, updated every 20 minutes. Timeseries consists of over 20,000 samples. (Bi) Raw data are in gray and random sampling excluding 12-day data drops are in orange. (Bii) Raw data are in gray and the reconstructed signal using model output based on input data with 12-day data drops is in orange. (C) Average complex wavelet transform (CWT) spectrum from the raw data in (A) is in gray. The BPWP spectral output based on the sampling in (Bi) input is shown in orange. Black stars denote significant peaks; peaks whose amplitude was above the 99[th] percentile of the distribution created by shuffling the input data and re-calculating BPWP 100 times. Data drops of thirty- and sixty- days duration, signal reconstructions, and method spectra are shown in (D) and (E) and in (F) and (G) respectively. The total number of samples for BPWP is fixed across the conditions at n = 1307 which is approximately 4 samples per day assuming no drops.
(TIF)

**S9 Fig. Subject-specific 10-fold 90/10 cross-validation for delta parameter selection.** Mean square error (MSE) between the real and estimated samples was calculated for each iteration of $\delta$ parameter and density of random sampling. Subplots (i), (ii), (iii) show results from real-world IED timeseries from participants 1, 2, and 3 respectively.
(TIF)

**S10 Fig. BPWP method performance, participant 2.** (A) Diagram depicting the approach to comparing the original IED rate timeseries (black) with the BPWP model output (orange). Each timeseries was bandpass filtered and the two filtered timeseries were used to calculate the Pearson correlation coefficient. (B) The Y axis shows the Pearson correlation coefficient for each bandpass filter with central periods listed on the X axis. P value for each point depicted was < 0.001. (C) Examples of filter outputs for the approximate daily (top), nineteen- (middle), and one hundred- day (bottom) cycles. The filtered original signal is in black, the filtered BPWP output is in orange. The insert describes the daily cycle filter outputs from 5 to 30 days.
(TIF)

**S11 Fig. BPWP method performance, participant 3.** (A) Diagram depicting the approach to comparing the original IED rate timeseries (black) with the BPWP model output (orange). Each timeseries was bandpass filtered and the two filtered timeseries were used to calculate the Pearson correlation coefficient. (B) The Y axis shows the Pearson correlation coefficient for each bandpass filter with central periods listed on the X axis. P value for each point depicted was < 0.001. (C) Examples of filter outputs for the approximate daily (top), nineteen- (middle), and one hundred- day (bottom) cycles. The filtered original signal is in black, the filtered BPWP output is in orange. The insert describes the daily cycle filter outputs from 5 to 30 days.
(TIF)

**S12 Fig. BPWP method output spectra compared with reference distribution built from 1,000 iterations.** Significant peaks in the output power spectra from the simulated data, participants 1, 2, and 3 are shown in sections (A), (B), (C), and (D) respectively. In i, the reference distribution for significance calculations was the result of re-running the model with re-shuffled input data 100 times. In ii, the model was re-run 1,000 times.
(TIF)

**S13 Fig. BPWP method output spectra compared with reference distribution built from 10,000 iterations.** Significant peaks in the output power spectra from the simulated data, participants 1, 2, and 3 are shown in sections (A), (B), (C), and (D) respectively. In i, the reference distribution for significance calculations was the result of re-running the model with re-shuffled input data 100 times. In ii, the model was re-run 10,000 times.
(TIF)

## Acknowledgments

We would like to give special thanks and our deepest appreciation to the participants for collecting these data. Additional thanks to Karla Crockett and Cindy Nelson for their vital clinical and administrative efforts in all aspects of this study. We acknowledge Medtronic for providing the Summit RC+S system and key scientific collaboration from Abbey Becker PhD, Dave Linde, and Rob Raike PhD, in particular.

## Author Contributions

**Conceptualization:** Irena Balzekas, Joshua Trzasko, Paul E. Croarkin, Gregory A. Worrell.

**Data curation:** Irena Balzekas, Filip Mivalt, Vladimir Sladky, Vaclav Kremen.

**Formal analysis:** Irena Balzekas, Joshua Trzasko, Thomas J. Richner, Nicholas M. Gregg, Vaclav Kremen.

**Funding acquisition:** Gregory A. Worrell.

**Investigation:** Irena Balzekas, Filip Mivalt, Vladimir Sladky, Jamie Van Gompel, Kai Miller, Paul E. Croarkin, Vaclav Kremen, Gregory A. Worrell.

**Methodology:** Irena Balzekas, Joshua Trzasko, Grace Yu, Thomas J. Richner, Vladimir Sladky, Vaclav Kremen, Gregory A. Worrell.

**Project administration:** Irena Balzekas, Filip Mivalt, Vaclav Kremen.

**Resources:** Irena Balzekas, Filip Mivalt, Vladimir Sladky, Paul E. Croarkin, Vaclav Kremen, Gregory A. Worrell.

**Software:** Irena Balzekas, Joshua Trzasko, Grace Yu, Thomas J. Richner, Filip Mivalt, Vladimir Sladky, Nicholas M. Gregg, Vaclav Kremen.

**Supervision:** Irena Balzekas, Joshua Trzasko, Paul E. Croarkin, Vaclav Kremen, Gregory A. Worrell.

**Validation:** Irena Balzekas, Grace Yu.

**Visualization:** Irena Balzekas, Joshua Trzasko, Thomas J. Richner, Filip Mivalt, Kai Miller, Paul E. Croarkin, Vaclav Kremen, Gregory A. Worrell.

**Writing – original draft:** Irena Balzekas, Joshua Trzasko, Grace Yu, Thomas J. Richner, Filip Mivalt, Vladimir Sladky, Nicholas M. Gregg, Jamie Van Gompel, Kai Miller, Paul E. Croarkin, Vaclav Kremen, Gregory A. Worrell.

**Writing – review & editing:** Irena Balzekas, Joshua Trzasko, Grace Yu, Thomas J. Richner, Filip Mivalt, Vladimir Sladky, Nicholas M. Gregg, Jamie Van Gompel, Kai Miller, Paul E. Croarkin, Vaclav Kremen, Gregory A. Worrell.

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
