## [Decision Letter · Decision Letter 0]

28 Jun 2023

Dear Ms Balzekas,

Thank you very much for submitting your manuscript "Method for cycle detection in sparse, irregularly sampled, long-term neuro-behavioral timeseries: Basis pursuit denoising with polynomial detrending of long-term, inter-ictal epileptiform activity" for consideration at PLOS Computational Biology. As with all papers reviewed by the journal, your manuscript was reviewed by members of the editorial board and by several independent reviewers. The reviewers appreciated the attention to an important topic. Based on the reviews, we are likely to accept this manuscript for publication, providing that you modify the manuscript according to the review recommendations.

Please review the reviewers main points on the robustness of the statistics and clearer reporting of the quantitative comparisons of the BPWP and CWT.

Sincerely,

Emma Claire Robinson

Academic Editor

PLOS Computational Biology

Thomas Serre

Section Editor

PLOS Computational Biology

Reviewer's Responses to Questions

**Comments to the Authors:**

Reviewer #1: This is a nice presentation.

Minor concern:

- recomputing 100 times and taking the 99th percentile does not make much sense to me. I would recommend recomputing 10000 or better yet 100000 times and using the 99th percentile. This leaves much less to randomness and gets a more clear boundary of the distribution. Unless it is the case that only 100 shufflings are sufficient to fully sample the distribution, which I suspect is not the case.

Reviewer #2: The paper introduces a novel method, termed Basis Pursuit-denoising With Polynomial Detrending (BPWP), for reconstructing cycles in sparse interictal spike event data. The authors thoroughly test the proposed approach on data collected from epileptic patients implanted with the Medtornic Summit RC+S. The methodology is evaluated using reconstructions from randomly downsampled data and data with large chunks dropped out to simulate interrupted communication. The authors compare their results with Complex Wavelet Transforms (CWT) of the original data and demonstrate a strong correlation between seizure frequency and reconstructed data using the top modes of their models.

While I do not have the skills to review the mathematics in depth or the motivation behind the algorithm proposed, I think the equations are represented in a way that the results could be accurately reconstructed and used by another research team.

Specific Comments:

1. The authors mention that the BPWP algorithm is computationally intensive and not suitable for real-time analysis on an embedded system. It would be helpful to include additional information on how the computational time scales with the length of data. This would provide insights into the scalability of the approach and potential limitations when dealing with larger datasets.

2. One major concern is that the peaks of BPWP shown in Figure 3F and 4F do not appear to align well with the CWT results, which exhibit only a few modes. While the paper states that the spectrum aligns visually with the main peaks in some examples and the methods section discusses quantitative comparisons between the two, I could not find these more quantitative results.

3. Figure 2 is not referenced in the main text. It is essential to include appropriate references to all figures within the text to guide readers and ensure proper understanding of the paper's content.

4. The mention of "figures 3 and 4" should be capitalized, as they represent figure titles.

5. In the discussion of the limitations of Classical Fourier Transform, the statement that it requires dense and regular sampling should be clarified. While Fast Fourier Transforms may have such requirements, the Fourier Transform itself is not inherently limited to regular sampling. It would be more accurate to state that the computation of sines and cosines for the Fourier Transform can be computationally intensive, making it less suitable for sparse time series.

6. Methods Section: The paper briefly mentions the hardware for extracting the Interictal Epileptiform Discharges (IEDs) but lacks a detailed explanation of the actual algorithm used. It is suggested to include a sentence or two describing the method employed, or at least provide a reference to the specific algorithm along with the hardware information. This addition would enhance the clarity of the methods section.

Overall, the paper presents an interesting methodology for reconstructing cycles in sparse interictal spike event data. The authors conduct thorough testing and analysis, providing evidence of robustness and accuracy. Addressing the specific concerns mentioned above, along with minor points, will significantly improve the clarity and completeness of the paper.

**Have the authors made all data and (if applicable) computational code underlying the findings in their manuscript fully available?**

Reviewer #1: Yes

Reviewer #2: Yes

PLOS authors have the option to publish the peer review history of their article (what does this mean?). If published, this will include your full peer review and any attached files.

Reviewer #1: No

Reviewer #2: **Yes: **Theoden I. Netoff

Figure Files:

Data Requirements:

Reproducibility:

References:

---

## [Decision Letter · Decision Letter 1]

4 Mar 2024

Dear Ms Balzekas,

We are pleased to inform you that your manuscript 'Method for cycle detection in sparse, irregularly sampled, long-term neuro-behavioral timeseries: Basis pursuit denoising with polynomial detrending of long-term, inter-ictal epileptiform activity' has been provisionally accepted for publication in PLOS Computational Biology.

Best regards,

Emma Claire Robinson

Academic Editor

PLOS Computational Biology

Thomas Serre

Section Editor

PLOS Computational Biology

Reviewer's Responses to Questions

**Comments to the Authors:**

Reviewer #1: While I continue to disagree with the idea that identifying the 99th percentile from 100 bootstrapped samples is appropriate, the authors have addressed my concern by empirically checking larger numbers of repeats. The well informed reader will still see this as a methodological error, but I am ok with publication now.

Reviewer #2: The authors have addressed all my concerns and I have no further concerns.

**Have the authors made all data and (if applicable) computational code underlying the findings in their manuscript fully available?**

Reviewer #1: Yes

Reviewer #2: Yes

PLOS authors have the option to publish the peer review history of their article (what does this mean?). If published, this will include your full peer review and any attached files.

Reviewer #1: No

Reviewer #2: **Yes: **Theoden Netoff

---

## [Editor Report · Acceptance letter]

4 Apr 2024

PCOMPBIOL-D-23-00707R1 

Method for cycle detection in sparse, irregularly sampled, long-term neuro-behavioral timeseries: Basis pursuit denoising with polynomial detrending of long-term, inter-ictal epileptiform activity

Dear Dr Balzekas,

I am pleased to inform you that your manuscript has been formally accepted for publication in PLOS Computational Biology. Your manuscript is now with our production department and you will be notified of the publication date in due course.

With kind regards,

Anita Estes
